# PtrARF2.1 Is Involved in Regulation of Leaf Development and Lignin Biosynthesis in Poplar Trees

**DOI:** 10.3390/ijms20174141

**Published:** 2019-08-24

**Authors:** Yongyao Fu, Papa Win, Huijuan Zhang, Chaofeng Li, Yun Shen, Fu He, Keming Luo

**Affiliations:** 1Chongqing Key Laboratory of Plant Resource Conservation and Germplasm Innovation, Institute of Resources Botany, School of Life Sciences, Southwest University, Chongqing 400715, China; 2Key Laboratory of Eco-Environments of Three Gorges Reservoir Region, Ministry of Education, Institute of Resources Botany, School of Life Sciences, Southwest University, Chongqing 400715, China; 3Department of Life Sciences and Technology, Yangtze Normal University, Fuling District, Chongqing 408100, China

**Keywords:** *Populus*, *P. trichocarpa* auxin response factor 2.1(PtrARF2.1), leaf morphology, lignin biosynthesis, transcription factor

## Abstract

Auxin response factors (ARFs) are important regulators modulating the expression of auxin-responsive genes in various biological processes in plants. In the *Populus* genome, a total of 39 ARF members have been identified, but their detailed functions are still unclear. In this study, six poplar auxin response factor 2 (PtrARF2) members were isolated from *P. trichocarpa.* Expression pattern analysis showed that *PtrARF2.1* is highly expressed in leaf tissues compared with other *PtrARF2* genes and significantly repressed by exogenous auxin treatment. PtrARF2.1 is a nuclear-localized protein without transcriptional activation activity. Knockdown of *PtrARF2.1* by RNA interference (RNAi) in poplars led to the dwarf plant, altered leaf shape, and reduced size of the leaf blade, while overexpression of *PtrARF2.1* resulted in a slight reduction in plant height and the similar leaf phenotype in contrast to the wildtype. Furthermore, histological staining analysis revealed an ectopic deposition of lignin in leaf veins and petioles of *PtrARF2.1-RNAi* lines. RNA-Seq analysis showed that 74 differential expression genes (DEGs) belonging to 12 transcription factor families, such as NAM, ATAF and CUC (NAC), v-myb avian myeloblastosis viral oncogene homolog (MYB), ethylene response factors (ERF) and basic helix–loop–helix (bHLH), were identified in *PtrARF2.1-RNAi* leaves and other 24 DEGs were associated with the lignin biosynthetic pathway. Altogether, the data indicate that PtrARF2.1 plays an important role in regulating leaf development and influences the lignin biosynthesis in poplars.

## 1. Introduction

In higher plants, leaf development is a dynamic and complex process thatrespondsto internal and external cues. Leaves are evolved from lateral branches and can be divided into two basic forms: simple and compound. A simple leaf has a single, continuous lamina, whereas a compound leaf contains multiple subunits termed leaflets, each resembling a simple leaf [1]. Several phytohormones, such as auxin (IAA), cytokinins (CK), gibberellins (GA) and jasmonic acid (JA), are involved in the leaf initiation and development [2]. It has been well established that auxin plays a critical role inleaf initiation [3], leaf serration [4] and phyllotaxis of leaf formation [5].

Leaves are derived from small populations of founder cells set aside on the flanks of the pluripotent shoot apical meristem (SAM), and auxin acts as a central regulator of this process. These are generated by auxin biosynthesis in the SAM and by directional auxin transport facilitated by the PIN-FORMED1 (PIN1) auxin transporter [6,7]. Inhibition of polar auxin transport by NPA or mutation of *PIN FORMED1 (PIN1)* partially represses the leaf initiation [8]. The YUCCA (YUC) family of flavin monooxygenases participates in converting IPA to indole-3-acetic acid and mutations of the genes of the *YUCCA* family also inhibit organ initiation [3]. Following initiation, the leaf primordia undergo elongation, morphogenesis and differentiation. It has been indicated that auxin and GA positively promote leaf dissection during leaf morphogenesis and lead to the initiated leaflets [9]. For example, the E3 ubiquitin-ligase BIG BROTHER (BB) can inhibit the growth of leaf lamina, probably by marking cellular proteins for degradation [10]. The pattern of leaf serration in *Arabidopsis* is dictated by the polar auxin transport (PAT) system in conjunction with the CUP-SHAPED COTYLEDON 2 (CUC2) [4]. Mutations of auxin efflux carriers *AtPIN1–7* produced the smooth leaves while over-expression of *AtPIN1* in this loss-of-function mutant generated serrate leaf margins [4]. In addition, the auxin importers (*AUX1/LAX* gene family) are also involved in the formation of the leaf margins by regulating the outputs of the PAT system. Mutation of three auxin importers (*aux1/lax1/lax2*) is required for delaying the serration growth [11].

In the past decades, remarkable advances have been made in understanding auxin signal transduction pathway [12,13,14]. It has been known that three transcription factor families, including the AUXIN/INDOLE-3-ACETIC ACID (Aux/IAA) transcriptional regulators, the TRANSPORT INHIBITOR RESISTANT 1/AUXIN SIGNALING F-BOX (TIR1/AFB) F-box proteins and sequence-specific binding proteins called AUXIN RESPONSE FACTORs (ARFs), are the key components of this pathway. Auxin promotes the interaction between TIR1/AFB and Aux/IAA, thereby triggering ubiquitin-mediated degradation of the Aux/IAA proteins via the proteasome [15,16]. The Aux/IAA and ARF proteins are involved in the typical pathway of auxin-dependent transcriptional regulation [17]. Aux/IAA transcription factors suppress the activity of ARF proteins, whereas their degradation results in the activation of ARF proteins and upregulation of subsequent auxin-responsive gene expression [18]. As one type of short-lived nuclear proteins, Aux/IAAs have been characterized to be transcriptional repressors of auxin response genes [19]. ARFs are capable of acting as transcriptional activators or repressors that regulate the expression of primary/early auxin responsive genes including members of *Gretchen Hagen 3* (*GH3*), *Small auxin up RNA* (*SAUR*) and *Auxin/indole acetic acid* (*Aux/IAA)* families by specifically binding to TGTCTC auxin-responsive cis-acting elements in the promoters of these genes [20].

Previous studies have identified that ARF genes play important roles in plant developmental processes, such as floral organ abscission, leaf development and senescence, fruit ripeningand lateral root formation [21,22,23,24]. For example, *Arabidopsis* auxin response factor 2 (AtARF2) acted as a transcriptional repressor in controlling leaf senescence [22]. Regulation of *Arabidopsis* auxin response factors AtARF3 and AtARF4 by *TAS3*-derived trans-acting short-interfering RNAs (tasiRNAs) affected the phyllotaxis and leaf shapes [25,26]. *Arabidopsis* auxin response factor 5 (MP/ARF5) was identified to be essential in determining the leaf initiation and vascular pattern formation [27,28]. In tomato, several ARFs, such as *Solanum lycopersicum* auxin response factors SlARF2A, SlARF4, SlARF7A, SlARF8B, have also been revealed to participate in the regulation of the leaf development [29]. Down-regulation of *S. lycopersicum* auxin response factor 4 (*SlARF4*) resulted in defective leaf phenotypes, including enhanced chlorophyll content and severe leaf curling shape [30]. Overexpression of *S. lycopersicum* auxin response factor 10A (*SlARF10A*) affected the auxin-driven processes, leading to the reduced lamina outgrowth [31]. To date, it is well known that *ARF* genes are involved in regulating leaf development in herbaceous plants, however, the detailed functions of these genes in woody species are still largely unknown.

With the release of the *P. trichocarpa* genome sequence [32], a total of 39 *ARF* genes were identified in the *Populus* genome (V1.0) [33]. Recently, Yang et al. (2014) [34] isolated 20 ARF genes related to adventitious root development from hybrid poplars. Transgenic plants overexpressing *PtrMP1/ARF5* displayed no obvious phenotypic differences compared with wild-type controls [35]. To investigate poplar *ARF* genes involved in regulating leaf development, we isolated the putative ortholog of *Arabidopsis ARF2*, *PtrARF2.1* and *PtrARF2.2* in this study, which are highly expressed in leaves compared with other *PtrARF2* genes, from *P. trichocarpa.* Although down-regulation of *PtrARF2.2* displayed slight changesin leaf phenotypes, transgenic plants carrying *PtrARF2.1-RNAi* displayed severe leaf phenotypes, such as irregular shapes and reduced size, while overexpression of *PtrARF2.1* did not affect leaf development. Moreover, our results revealed an ectopic deposition of lignin in the leaf veins and petioles of *PtrARF2.1-RNAi* lines. These results indicated that PtrARF2.1 is an essential regulator in leaf development and influences lignin biosynthesis in poplar.

## 2. Results

### 2.1. Identification and Characterization of Poplar ARF2 Transcription Factors

A previous study has shown that at least six *ARF2* genes were found in the genome of *P. trichocarpa* V1.0 [33]. Through blast research and using ARF2 protein domains in the genome of *P. trichocarpa* V3.0, six *ARF2* genes were also confirmed. These *PtrARF2* genes were mapped into six different chromosomes and they displayed the conserved genomic structures with regard to the numbers and positions of introns and exonsexcept for *PtrARF2.3* (Appendix A). These PtrARF2 proteins except for PtrARF2.3, which is lack of an Aux/IAA domain responsible for binding with IAA factors—have three typical functional domains (DNA-binding domain/B3 type, DBD/B3, an activation or repression domain, ARF and IAA dimerization domain, Aux/IAA). It has been revealed that AtARF2 factor is a pleiotropic developmental regulator and modulates the leaf development and senescence [21,36]. Pairwise comparison of amino acid sequence revealed that PtrARF2.1 shared 65.2% identity with AtARF2 (Figure 1A).

To investigate phylogenetic relationship among ARF2 proteins in herbaceous and woody species, a phylogenetic tree was constructed on the basis of amino acid sequences by the Neighbor-Joining methods (Figure 1B). Obviously, all the ARF2 proteins fall into two classes (I, II) and Class I can be further subdivided into Subclass Ia and Subclass Ib. PtrARF2.1 and PtrARF2.2 are grouped together with *Arabidopsis* AtARF2 and *Eucalyptusgrandis*auxin response factor 2A (EgrARF2A), *Solanum lycopersicum* auxin response factor 2A (SlARF2a) and auxin response factor 2B (SlARF2B) belonging to Subclass Ia, suggesting that PtrARF2.1 and PtrARF2.2 are potential transcriptional regulators involved in the leaf development. PtrARF2.3 and PtrARF2.4, close to *Vitis vinifera* auxin response factor 2 (VvARF2) are clustered into Subclass II, while PtrARF2.5 and PtrARF2.6 share highly homology with *Eucalyptus grandis* auxin response factor 2B (EgrARF2B), belonging to a new clade (Ib) found preferentially in woody plants.

### 2.2. Expression Profiles of PtrARF2 Genes in Poplar Tissues

We determined their expression patterns in different tissues of poplar by quantitative reverse transcription polymerase chain reaction (qRT-PCR). The results revealed that *PtrARF2.1* and *PtrARF2.2* were expressed in all tissues tested, whereas no expression was detected for *PtrARF2.3* and a low expression for *PtrARF2.4* and *PtrARF2.6*. Additionally, transcript level of *PtrARF2.5* was mainly expressed in roots and stems (Figure 2A). Furthermore, we identified that *PtrARF2.1* is predominantly expressed in poplar leaves in contrast to other *PtrARF2* homologsby semi-quantitativePCR analysis (Figure 2B).

Previous studies have shown that the expression of *AtARF2* and *SlARF2A,B* in developmental processes are regulated by auxin signaling [37,38]. We found some conserved auxin-related cis-acting regulatory elements in the promoter regions of *PtrARF2.1*.qRT-PCR analysis revealed that transcript accumulation of *PtrARF2.1* was significantly repressed by exogenous auxin at 3 h (Appendix A). Additionally, *PtrIAA3.1* of *AtIAA3* homolog known to be regulated by auxin was used as the control to validate the efficacy of auxin treatment.

### 2.3. PtrARF2.1 Localizes to the Nucleus and Is Not a Transcription Activator

To investigate the localization of PtrARF2.1 in vivo, the recombinant construct of the *PtrARF2.1* fused to *RFP* (Red fluorescent protein) reporter gene driven by the cauliflower mosaic virus (CaMV) 35S promoter was transformed into tobacco epidermal cells via particle bombardment method. As shown in Appendix A, the fusion protein was found to accumulate exclusively in the nucleus while the control *35S::RFP* was distributed throughout the whole cell, in accordance with the function of PtrARF2.1 as a transcription factor.

To explore whether PtrARF2.1 can activate early auxin response, the full-length ORF of *PtrARF2.1* was fused with the GAL4 DNA-binding domain coding sequence and then transformed into the yeast strain Y2HGold. The yeast transformants harboring the constructof *GAL4:PtrARF2.1* fusion could not grow on the selection medium lacking adenine (A), histidine (H) and tryptophan (T). While the control cells carrying *GAL4:VP16* fusion gene grew well and also turned to be blue induced by the expression of the *b-galactosidase* reporter gene (Appendix A), suggesting that PtrARF2.1 has not been a transcriptional activator.

### 2.4. Exploring the Role of PtrARF2.1 through a Reverse Genetics Approach

To investigate biological role of PtrARF2.1 protein, transgenic poplar lines under- and overexpressing *PtrARF2.1* were acquired in the *P. tomentosa* genetic background. To this purpose, dedicated RNAi construct was designed to target only *PtrARF2.1* while not other *PtrARF2* genes. Following PCR analysis of *hptII* gene in under- and over-regulated transgenic plants using genomic DNA, expression analysis by qRT-PCR indicated that transcript level of *PtoARF2.1*, the paralog of *PtrARF2.1* in *P. tomentosa* was specifically reduced in *PtrARF2.1-RNAi* and *PtrARF2.1* was strikingly upregulated in overexpression of *PtrARF2.1* (*PtrARF2.1*-OE) lines, respectively (Figure 3C). It is important to note that the expression of the most closely related *PtrARF2* genes except *PtoARF2.1* in terms of sequence identity was not changed in transgenic lines compared to the control. However, *PtrARF2.3* mRNA levels were below the detection limit (Appendix A), thus ruling out a lack of specificity of the RNAi strategy.

It is expected that *PtrARF2.1-RNAi* lines displayed a visible phenotype related to leaf morphology, including the asymmetric leaf blades, the serrate leaf margins. and the reduced area of leaves (Figure 3D,E, Appendix A). To allow for a quantitative comparison of leaf shape in different genetic backgrounds, an important parameter, compactness (a measure of the ratio of circumference to area) was tested, which is especially useful for discriminating 2D leaf shape according to Kasprzewska et al. (2015) [11]. As shown in Figure 3F, transgenic leaves of *PtrARF2.1-RNAi* had significantly higher values for this parameter compared to wild-type (WT) plants, suggesting that PtrARF2.1 is an essential regulator in leaf development in poplars. In addition, the RNAi lines displayed a dwarf phenotype with a strongly inhibited vegetative growth in the greenhouse (Figure 3A). Whereas *PtrARF2.1*-OE lines showed less reduced vegetative growth and the same leaf phenotypes compared with the WT control (Figure 3A,D).

### 2.5. Silencing of PtrARF2.1 Causes Ectopic Deposition of Lignin in Poplar Leaves

Previously, our studies showed that overexpressing *PtoMYB92* or *PtrMYB152* resulted in the changed phenotype of poplar leaves, which reminded us that PtrARF2.1 most likely influences the biosynthesis of lignin as like PtoMYB92 or PtrMYB152 in leaf tissues [39,40]. To explore the role of PtrARF2.1, histochemical staining was performed for leaf tissues, which disclosed an enhanced lignin deposition in leaf veins of *PtrARF2.1-RNAi* but not in that of *PtrARF2.1*-OE lines (Figure 4A). Furthermore, confocal microscopy of lignin autofluorensence was performed to test these veins, which displayed strong signals of lignin autofluorescence in *PtrARF2.1-RNAi* leaves (Figure 4B). The anatomical cross-sections of poplar petioles were also used for the histological analysis. The results further revealed an ectopic deposition of lignin in *PtrARF2.1-RNAi* lines compared to that in *PtrARF2.1*-OE or the WT (Figure 4A). To quantitatively test the level of lignification, total of lignin content in *PtrARF2.1* suppressed leaves was measured by the acetyl bromide (AcBr) method. The lignin content in *PtrARF2.1-RNAi* leaves was almost twofold higher than that of the WT control (see Appendix A). Similar result was found in the stems of *PtrARF2.1-RNAi* lines. Altogether, knock-down of *PtrARF2.1* led to ectopic deposition of lignin in poplar leaves.

### 2.6. RNA-Seq Transcriptome Analysis of the PtrARF2.1 RNAi Line

To identify the different expression genes (DEGs) in the *PtrARF2.1* suppressed lines, RNA-Seq using Illumina sequencing technology was performed with two biological replicates and the results showed to be highly consistent. RNA-Seq analysis totally generated 24,106,910 and 24,129,551 clean reads for *PtrARF2.1-RNAi* Line 1 and WT (*P. tomentosa*), which was about 50 bp in length. The reads were aligned to the poplar reference genome database using HISAT and Bowtie2 software. Of the total reads, 84.92% and 85.06% were matched to genomic locations in the *PtrARF2.1* suppressed and WT plants, respectively. A total of 2042 genes were identified asstatistically significant (false discovery rate [FDR] <0.05) between the suppressed *PtrARF2.1* and WT plants (see Appendix A). There were 1429 upregulated genes and 613 downregulated genes in *PtrARF2.1-RNAi* Line 1 compared with WT. The functions of the DEGs were further identified using GO analysis. All the DEGs were categorized into three main categories including biological process, cellular component, and molecular function, in which there were 20, 12, and 11 functional groups, respectively (Figure 5).

Furthermore, DEGs coding transcription factors (TFs) were analyzed in the RNA-Seq database, which revealed 74 DEGs belonging to 12 TF families (see Appendix A). The families including more DEGs areMYB (19 DEGs), NAC (14 DEGs), ERF (10 DEGs) and bHLH (7 DEGs). Notably, the DEGs in most of TF families were upregulated except that in WRKY (transcription factor containing the WRKY domain and zinc-finger-like motif), SRF-TF(serum response factor) and B3 (transcription factor containing a B3 domain) (Figure 6). Besides, four TF families including GATA (zinc-finger transcription factor), Ovate, Trihelix and ZF (transcription factor containing a zinc finger homeodomain) were totally upregulated in *PtrARF2.1-RNAi* Line 1. These results indicated an important role of TFs in transcriptional regulation of leaf development, suggesting that PtrARF2.1 may function as a negative regulator in poplars.

### 2.7. Expression Analysisof Lignin Metabolic Genes in PtrARF2.1 Suppressed Lines

To investigate if PtrARF2.1 was involved in the lignin biosynthesis, we searched the RNA-seq data and found thata total of 24 genes related to the lignin metabolic process were induced in *PtrARF2.1* suppressed Line 1 (see Appendix A), in which 16 DEGs were involved in the phenylpropanoid/monolignol biosynthetic pathway, such as 4-coumarate:CoA ligase gene (*4CL5*), cinnamate 4-hydroxylasegene (*C4H2*), cinnamyl alcohol dehydrogenasegene (*CAD1*), ferulic acid 5-hydroxylasegene (*F5H2*), laccase genes (*LACs*) and so on. Additionally, eight DEGs encoded for NACs and MYBs, which worked as master switches in wood formation of poplars [41,42]. To better understand the molecular network regulated by PtrARF2.1, key genes in the lignin biosynthesis were further tested by qRT-PCR analysis. As shown, most of genes involved in the phenylpropanoid/monolignol biosynthetic pathway were upregulated in *PtrARF2.1* suppressed lines, such as phenylalanine ammonia-lyasegene (*PAL4*), *C4H2*,4-coumaroylshikimate 3-hydroxylasegene (*C3H3*), *CAD1* and *F5H2*, while some were down-regulated in *PtrARF2.1* over-expressed lines (Figure 7A). Transcription factors, including wood-associated NAC domain transcription factor 2B (*WND2B*), *MYB020*, *MYB92* except for *MYB152*were also induced in *PtrARF2.1*-RNAi lines while *WND2B*and *MYB92*repressed in *PtrARF2.1* overexpressed lines (Figure 7B). In summary, PtrARF2.1 is an important regulator implicated in the lignin biosynthesis of poplar leaves.

## 3. Discussion

The ARF family of transcription factors play critical roles in a variety of auxin-mediated plant growth and development processes. Since the report describing ARF families from the herb plants *Arabidopsis* and *Tomato* [29,43], this family of 39 ARF members has been identified from the woody tree using the released genomic data of *P. trichocarpa* V1.0 [33]. Following the isolation of expression analysis of 20 ARF genes from the poplar roots [34], the present study provides new insights on PtrARF2.1, one member of PtrARF2 genes, based on the latest data of the *P. trichocarpa* genome V3.0 (https://phytozome.jgi.doe.gov/pz/#!info?alias=Org_ Ptrichocarpa). Except for PtrARF2.3, the other members have three functional domains of a typical ARF protein. In addition, PtrARF2.3 is probably a pseudogene with the first amino acid (N). By contrast, threenew PtrARF2 members are found in the *P. trichocarpa* genome V3.0. However, PtrARF2.1 and PtrARF2.2 are the closest to *Arabidopsis* AtARF2 and tomato SlARF2A/B in poplars.

Our understanding of the functional significance of ARF proteins has come mainly through the study of loss-of-function *arf Arabidopsis* mutant lines. Utilizing T-DNA insertion lines, AtARF2 was suggested to play important roles in multiple developmental processes, such as cell division, organ growth, leaf senescence, flowering and floral organ abscission [21,22,36,38]. Herein, we reported on the role of PtrARF2.1 through a reverse genetics approach in *Populus*. Up- and/or downregulation of *PtrARF2.1* led to reduced plant growth, especially in *PtrARF2.1-RNAi* lines that displayed the dwarf plants. Interestingly, overexpression of *PtrARF2.1* also resulted in slightly stunted the plant growth (Figure 3A). We speculated that PtrARF2.1 might be involved in auxin homeostasis, resulting in repressing the development of transgenic plants. It was also was noted that the knockdown of *PtrARF2.1* resulted in a severely morphological and developmental phenotype of the leaf tissue of transgenic lines, in contrast to that of the wide-type and *PtrARF2.1*-OE lines. This phenotype was consistent with the expression pattern of *PtrARF2.1* and indicative of a role for the encoded protein in poplar leaves.

What is the role of PtrARF2 in auxin signaling? ARFs with a glutamine-rich middle region (MR) usually act as activators of early auxin-responsive genes, while ARFs with proline- or/and serine-rich MRs or no specific amino acid-rich MRs as repressors. For example, AtARF2 inhibited the transcription of reporter genes under the control of synthetic AuxREs [20,44]. In previous reports, it was observed the *ore14/arf2 Arabidopsis* mutant exhibited enhanced sensitivity to auxin, and downregulation of *SlARF2* in tomato led to induced expression of auxin responsive genes [21,37]. As shown, *PtrARF2.1* gene was early repressed in response to auxin and the encoded protein had no transcriptional activation activity, suggesting that PtrARF2.1 is not arequirement for activating early auxin responsive genes. Moreover, the RNA-Seq data indicated the enhanced expression of auxin responsive genes in *PtrARF2.1*-inhibited lines. The combined evidence implies that PtrARF2.1 might function as a transcription repressor of auxin-dependent gene transcription, which is in accordance with the characterization of AtARF2 and SlARF2A/B. The results seen here also suggest that ARF2-mediated auxin signaling is conserved between the annual herbaceous and perennial woody plants.

Which ARF factor is the major player in leaf development? As is well-known, ARFs regulate auxin-mediated transcriptional activation or repression, and each ARF protein is thought to play a specific or central role in the developmental process of different tissues in plants. Through studies of loss-of-function or/and dominant gain-of-function *arf* mutants in *Arabidopsis*, AtARF2, ETT/ARF3, ARF4 and MP/ARF5 have shown to be critical regulators in the leaf development and morphogenesis [28,36,45,46,47]. A few lines of our data here, together with previous results, indicated that PtrARF2.1 might be an important player in leaf development in woody plants. Firstly, *PtrARF2.1* was preferentially expressed in leaf tissue of the *Populus* related to other *PtrARF2* genes. In addition, down-regulation of *PtrARF2.1* displayed severely irregular shape of leaves and the similar feature of leaf senescence with *ore14/arf2* mutant in *Arabidopsis* [21]. By contrast, over-expressing *PtrMP1/ARF5*, the homolog of AtARF5 did not change any phenotypes in poplars, although they induced a 2–4 fold increase in the expression of a *Populus* homolog of *AtHB8* [35,48].

Interestingly, the presented data revealed that suppression of *PtrARF2.1* resulted in an ectopic deposition of lignin in poplar leaves. RNAi-Seq analysis showed that a lot of genes involved in the lignin biosynthesis were activated in *PtrARF2.1-RNAi* Line 1. For example, *P. trichocarpa* laccase (*PtrLAC*) genes had been pointed out to be targeted by PtrARF2 factor in an early study [49], suggesting that PtrARF2.1 probably works as a molecular switch implicated in the upstream of the phenylpropanoid/monolignol biosynthetic pathway. In addition, transcript abundance of *WND2B*, *WND6B* and *MYB92* were upregulated in *PtrARF2.1-RNAi* leaves, which reported to be positive regulators involved in lignin biosynthesis [40,41]. However, the transcript level of *PtrMYB152* was inhibited, which induced the expression of secondary wall biosynthetic genes [39]. To better understand the molecular mechanism of PtrARF2.1, qRT-PCR analysis further indicated that lignin biosynthetic genes *PAL4*, *PtrC4H2*, *CAD1*, etc. and lignin-associated *TFsWND2B*, *MYB020* and *MYB92* were up-regulated in *PtrARF2.1-RNAi* lines whilesomedownregulated in *PtrARF2.1-OE* lines, implying that they might be directly targeted by PtrARF2.1. Taken together, the present work summarized that PtrARF2.1 plays an important role in the leaf development and regulated the lignin biosynthesis.

## 4. Materials and Methods

### 4.1. Plant Materials and Growth Conditions

*Populus trichocarpa* Torr. & A. Grayand *P. tomentosa* Carr. (clone 73) was grown in the greenhouse at 25 °C under a 16/8-h photoperiod with supplemented light (4500 lux), and around 60% relative humidity for optimum growth.

### 4.2. Sequence Alignment and Phylogenetic Analysis

BLAST analysis was performed at the plant genomics resource website (https://phytozome.jgi.doe.gov/pz/#!search?show=BLAST) using by AtARF2 sequence. Multiple sequence alignments were performed using DNAMAN software. Phylogenetic relationships of ARF2 proteins from *P. trichocarpa* and other species were constructed by the neighbor-joining method using MEGA version 5.05 [50].

### 4.3. Cloning of PtrARF2.1

The cDNA fragments encoding PtrARF2.1 were amplified with gene-specific primers (Appendix A) based on the sequences of Potri.012G106100.v3.0 from *P. trichocarpa* by polymerase chain reaction (PCR). The PCR reaction was carried out with pfu DNA polymerase (Takara, Dalian, China) in a total volume of 50μL at 94 °C for 3 min; 35 cycles of 94 °C for 30 s, 58°C for 45 s and 72 °C for 75 s, followed by a final extension of 72 °C for 10 min. The PCR product was cloned into the pMD20-T Vector (Takara, Dalian, China) and confirmed by sequencing (BGI, Beijing, China).

### 4.4. Auxin Treatment for Poplar Tissues

For auxin treatment, 18-day-old poplar seedlings were soaked in liquid 1/2 MS medium with or without (mock treatment) 20 μM IAA for 3 h, 6 h, 12 h and 24 h. Then, all the seedling tissues were immediately frozen in liquid nitrogen and stored at −80 °C until RNA extraction.

### 4.5. Yeast One-Hybrid Assay

The full-length ORF of *PtrARF2.1* was amplified with specific primers (Appendix A), the amplification product was inserted into pGBKT7 (Clontech) using ClonExpress II (Vazyme Biotech, Nanjing, China) and the recombinant plasmid was introduced into the yeast strain *Saccharomyces cerevisiae* Gold2 by the method described previously (Zaragoza et al. 2004). Transformants were grown on SD medium lacking Trp (tryptophan) for selection of positive clones and then on SD medium lacking Trp, His (histidine) and Ade (adenine) for the transactivation assay. The X-α-gal (Solarbio, Beijing, China) was used to identify the transcription activation activity of PtrARF2.1.

### 4.6. Transformation of Poplar Plants

To construct the RNAi vector, the specific sequence (260 bp) of *PtrARF2.1* was obtained by reverse transcription-PCR (RT-PCR) using gene-specific primers (see Appendix A). The PCR products were fused around a spacer of GUS fragment by over-lap PCR and then cloned into the *Bam*HI site of the plant binary vector pCXSN [51] driven by the Cauliflower mosaic virus (CaMV) 35S promoter. For the overexpression vector, the full-length ORF of *PtrARF2.1* was amplified by RT-PCR using gene-specific primers (Appendix A) and ligated into the *Bam*HI site of the pCXSN vector. The resulting vectors *p35S::PtrARF2.1-RNAi* and *p35S::PtrARF2.1-*OE were transferred into *Agrobacterium tumefaciens* strain GV3101 by the freeze-thaw method, respectively.

Transgenic poplar plants were generated via *A. tumefaciens-*mediated transformation as described previously by Jia et al. (2010) [52]. In brief, leaves of Chinesewhite poplar (*P. tomentosa* Carr.) were infected by recombinant *Agrobacterium* and putative transgenic shoots were selected on WPM medium supplemented with 9 mg/L hygromycin. Rooted plantlets were acclimatized in pots at 25 °C in a 16-h photoperiod with 60% relative humidity for threeweeks and then transferred to the greenhouse for further studies.

### 4.7. RNA Extraction and Quantitative Real-Time PCR

Total RNA was extracted from several fresh tissues of poplar plants using by plant Trizol Reagent (Tiangen, Beijing, China) or Biospin Plant Total RNA Extraction Kit (Bioer, Hangzhou, China) according to the manufacturer’s instructions. First-strand cDNA was synthesized from 2 μg RNA using a PrimeScript^TM^RT reagent Kit (Takara, Dalian, China) in a total volume of 20 μL by using oligo dT_18_ at 42 °C for 30 min. Quantitative real-time PCR (qRT-PCR) was performed as described by Li et al. (2015) in a 20-μL reaction volume containing 10 μL of SYBR Premix Ex Taq II (TliRNaseHPlus,Takara, Dalian, China), 5–10 fold dilution of cDNAs and gene-specific primers (Appendix A), which were designed using QuantPrime online. The expression of *18S rRNA* was used as reference for calculating the relative amount of target gene expression using the 2^ΔΔCT^method [53]. qRT-PCR analysis was based on at least three biological replicates for each sample with three technical replicates.

### 4.8. RNA-Seq Analysis

RNA-Seq was carried out at BGI’s Technology Co., Ltd. (Wuhan, China). Total RNA was extracted from fresh leaves using the plant Trizol Reagent (Tiangen, Beijing, China) and the RNA quality was determined by the NanoDrop and Agilent 2100 bioanalyzer. cDNA libraries were constructed by using a TruSeq™ RNA Sample Preparation Kit (Illumina) and subsequently sequenced using Illumina HiSeq™ 2000 platform. RSeQC-2.3.2 program (http://code.google.com/p/rseqc/) was used to assess the quality of RNA-Seq data. All clean reads were aligned to the *P. trichocarpa* v3.0genome (https://phytozome.jgi.doe.gov/pz/#!bulk?org=Org_Ptrichocarpa).

To determine differentially expressed genes (DEGs), transcript abundance of each gene was normalized by the fragments per kilobase of exon per million mapped reads (FRKM) method using Cuffdiff software (http://cufflinks.cbcb.umd.edu/). False discovery rate (FDR) ≤0.05 control method was used to identify the threshold for DEGs. Gene-annotation functional enrichment and KEGG pathway analysis was performed using GOATOOLS (https://github.com/tanghaibao/goatools) and KOBAS (http://kobas.cbi.pku.edu.cn/home.do).

### 4.9. Histology and AcBr Lignin Assay

Leaf blades and petioles of five-month-old plants grown in greenhouse were hand-cut with a razor blade or by an Ultra-Thin Semiautomatic Microtome (FINESSE 325, Thermo, Shanghai, China). The leaf blades and the microsections were stained for 30 s with 1.0% (Weight/Volume, *w*/*v*) phloroglucinol after dissociation for 60 s by 40% (Volume/Volume, *v*/*v*) HCl (hydrochloric acid), and detected under the microscope (BX53, Olympus, Tokyo, Japan).

Quantitativeanalysisoflignin was performed by the modified AcBr-soluble lignin method as described by Fukushima and Hatfield (2001) [54]. In brief, samples were dried at 65 °C for overnight and ground to pass a 40-mesh screen in a mill. The samples (50 mg) were resolved in 25% (*v*/*v*) acetyl bromide in acetic acid (5 mL) and put at 50 °C for 2 h treatment. After cooling, 10% of the solution was supplemented with 2.5 mL of sodium hydroxide (2 M), 0.35 mL of hydroxylamine hydrochloride (0.5 M) and 2.4 mL of glacial acetic acid. Finally, lignin content was measured by the ultravioletabsorbance at 280 nm according to Iiyama and Wallis (1988) [55].

### 4.10. Determination of Lignin Autofluorescence

The leaf blades and petioles of five-month-old poplars were hand-cut by a razor blade. Samples were fixed using a double-sided sticky tape and directly observed under a confocal laser microscope (FV3000, Olympus, Tokyo, Japan) following the manual’s recommendations.

### 4.11. Accession Numbers

The sequences of genes used in this study can be found in the Phytozome (V11.0) under the following numbers: AtARF2 (AT5G62000.1), SlARF2A (Solyc03g118290), SlARF2B (Solyc12g042070), VvARF2 (GSVIVT01011008001), EgrARF2A (Eucgr.K02197.1), EgrARF2B (Eucgr.B03551.1), PtrARF2.1 (Potri.012G106100.1), PtrARF2.2 (Potri.015G105300.1), PtrARF2.3 (Potri.002G207100.1), PtrARF2.4 (Potri.014G135300.1), PtrARF2.5 (Potri.001G066200.1), PtrARF2.6 (Potri.003G163600.1), PtrIAA3.1 (Potri.005G053800.1), PtrPAL4 (Potri.010G224100.1), PtrC4H2 (Potri.013G157900.1), Ptr4CL5 (Potri.003G188500.1), PtrHCT1 (Potri.003G183900.1), PtrC3H3 (Potri.006G033300.1), PtrCAD1 (Potri.009G095800.1), PtrF5H2 (Potri.007G016400.1), PtrLAC9 (Potri.004G156400.1), PtrLAC17 (Potri.008G073700.1), PtrLAC28 (Potri.010G193100.1), PtrWND2B (Potri.002G178700.1), PtrMYB003 (Potri.001G267300.1), PtrMYB020 (Potri.009G061500.1), PtrMYB92 (Potri.001G118800.1), PtrMYB152 (Potri.017G130300.1).

## 5. Conclusions

In the current study, we cloned an Auxin Response Factor PtrARF2.1, the homolog of AtARF2 from wood plants and revealed the functional significance of PtrARF2.1 in the leaf development through a reverse transgenics approach. Results presented here indicated that PtrARF2.1 is an essential regulator in the leaf development and influences the lignin biosynthesis. However, the molecular mechanism controlled by PtrARF2.1 remains unclear in wood plants. Future work is needed to betterunderstand how PtrARF2.1 participates in regulating the leaf formation and the lignin biosynthesis.

## Figures and Tables

**Figure 1 ijms-20-04141-f001:**
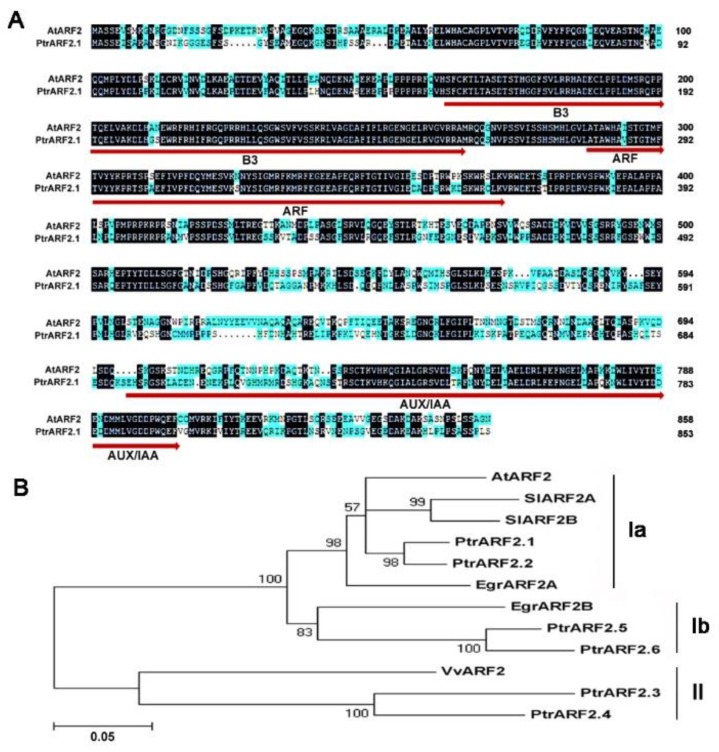
Amino acid sequence from *Populus trichocarpa* with other species. (**A**) Sequence alignment of PtrARF2.1 in *P. trichocarpa* with AtARF2 in *Arabidopsis thaliana*. Peptide sequences were aligned by DNAMAN software. Identical and similar amino acid residues are indicated by dark and blue background, respectively. Numbers on the right refer to amino acid sequence positions. These conserved DBD/B3, ARF and Aux/IAA domains in ARF2 protein are underlined. (**B**) Phylogenetic analysis of the poplar and selected ARF2 proteins was performed by the neighbour-joining method using MEGA version 5. Bootstrap support is indicated at each node. Bar, 0.05 substitutions per site. Additional sequences include *A. thaliana* (AtARF2), *Solanum lycopersicum* (SlARF2A and SlARF2B), *Vitis vinifera* (VvARF2), *Eucalyptus grandis* (EgrARF2A and EgrARF2B). The accession numbers of ARF factor sequences are given in the Materials and Methods section.

**Figure 2 ijms-20-04141-f002:**
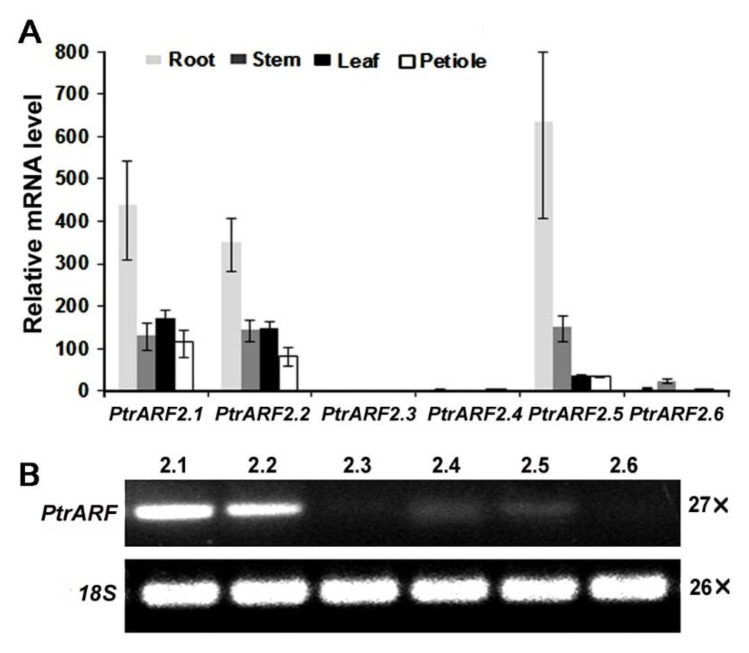
Expression patterns of *PtrARF2* genes in poplars. (**A**) Quantitative reverse transcription PCR analysis of all the *PtrARF2* genes in different organs and tissues of poplars. Transcript level of*PtrARF2.4* in different tissues was used as reference (relative mRNA level 1). Error bars mean ± SD of three biological replicates. (**B**) Semi-quantitative PCR analysis of *PtrARF2* genes in the leaf tissue. The poplar *18S* rRNA was used as an internal control.

**Figure 3 ijms-20-04141-f003:**
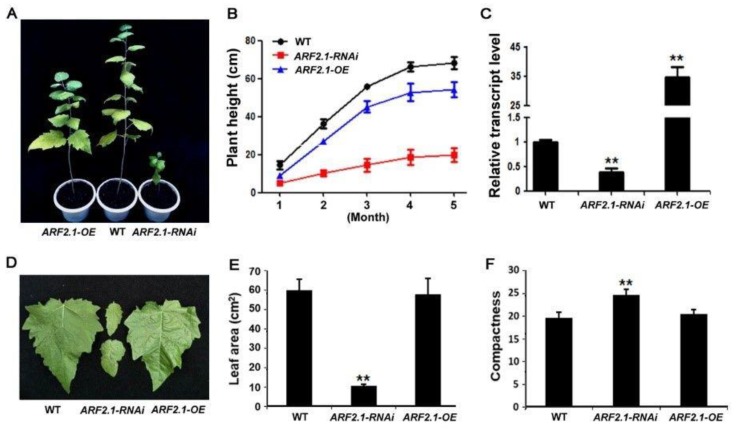
Phenotype analysis of *PtrARF2.1* overexpressed and suppressed poplars. (**A**) Representative five-month-old *PtrARF2.1*-OEand *PtrARF2.1-RNAi* plants compared with the wildtype (WT). (**B**) Comparison of plant height in WT, *PtrARF2.1*-OEand *PtrARF2.1-RNAi* plants. Measurements were made for 30 plants of each line during five months of plant growth in the greenhouse. Error bars mean ± SD value for each line. (**C**) Quantitative RT-PCR analysis of *PtrARF2.1* transcripts in representative *PtrARF2.1-RNAi* and *PtrARF2.1*-OE lines. The poplar *18S* rRNA was used as an internal control. Error bars mean ± SD of three biological replicates. (**D**) Representative 5-month-old leaf phenotypes from *PtrARF2.1-RNAi* and *PtrARF2.1*-OE lines in comparison with the WT control. (**E**) Comparison of the leaf area in *PtrARF2.1-RNAi*, *PtrARF2.1*-OE and WT plants. Error bars mean ± SD of three biological replicates. (**F**) Quantitative analysis of the leaf index value of compactness from *PtrARF2.1-RNAi*, *PtrARF2.1*-OE and WT plants. Error bars mean ± SD of three biological replicates. Stars indicate the statistical significance using Student’s *t*-test: ** *p*-value < 0.01.

**Figure 4 ijms-20-04141-f004:**
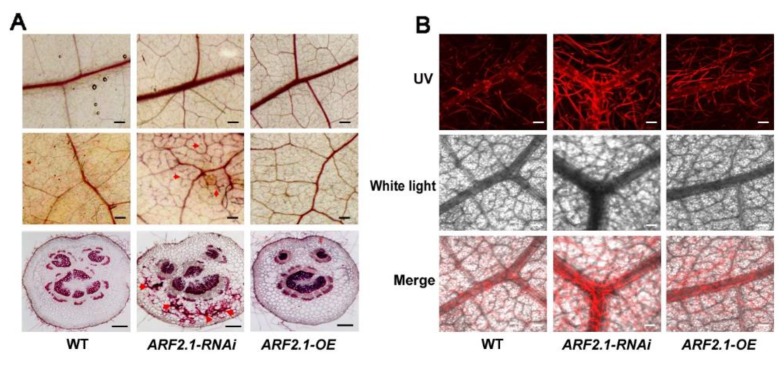
Effects of up- and downregulation of *PtrARF2.1* on the biosynthesis of lignin in poplar leaves. (**A**) The phloroglucinol-HCl analysis of lignin deposition in the main veins (top) and the fine veins (middle) and the petioles (bottom) of the leaf tissue from the WT, *PtrARF2.1-RNAi* and *PtrARF2.1*-OE (right). (**B**) Close-up view of the lignin signals in very fine veins in leaf blades of the WT (left) and *PtrARF2.1-RNAi* (right) poplars by laser scanning confocal microscopy. Scale bars: A (up) = 1.25 mm, A (middle) = 0.5 mm, A (down) = 200 μm; B = 150 μm.

**Figure 5 ijms-20-04141-f005:**
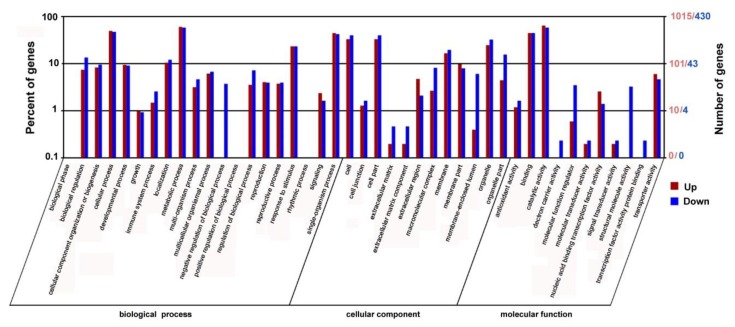
Functional analysis of the DEGs in *PtrARF2.1-RNAi* and WT plants based on the RNA-Seq data. GO functional enrichment analysis of DEGs between the *PtrARF2.1* down-regulated and WT plants based on sequence homology. The top X-axis refers to the number genes annotated by GO term. The bottom X-axis refers to classification of GO including Biological Process, Cellular Component and Molecular Function. The Y-axis represents the percentages of annotated genes accounted for the total annotated genes.The red bar represents upregulated genes and the blue bar represents downregulated genes.

**Figure 6 ijms-20-04141-f006:**
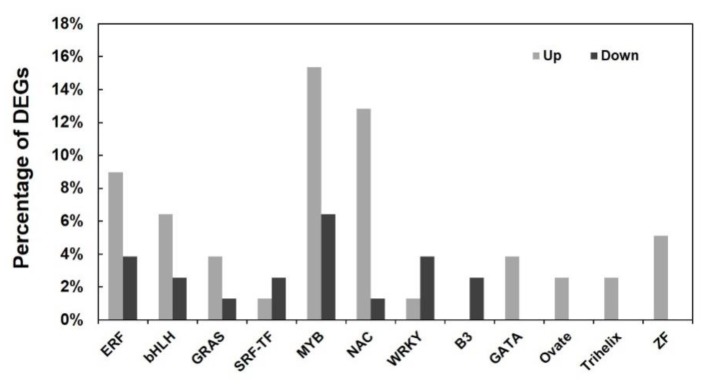
Up- or downregulation of DEGs encoding the members of transcription factor (TF) families in *PtrARF2.1* suppressed plants.

**Figure 7 ijms-20-04141-f007:**
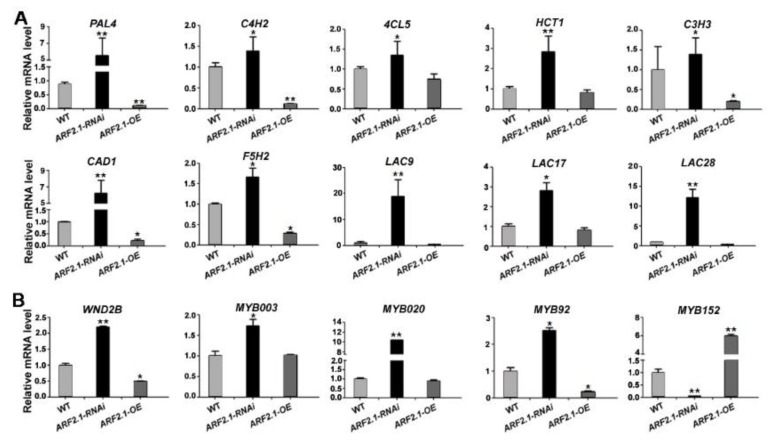
Relative expression levels of genes associated with the lignin biosynthesis in *PtrARF2.1-RNAi* and *PtrARF2.1*-OEpoplars. (**A**) Analysis of the expression levels of genes involved in the phenylpropanoid/monolignol biosynthetic pathway by qRT-PCR methods. (**B**) Analysis of the expression levels of the lignin-associated with *TF* genes by qRT-PCR methods. Total RNA was extracted from the WT, *PtrARF2.1-RNAi* and *PtrARF2.1*-OE leaves. The poplar *18S* rRNA was used as an internal control. Error bars mean ± SD of three biological replicates. Stars indicate the statistical significance using Student’s *t*-test: * *p*-value < 0.05, ** *p*-value < 0.01.

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
