# Peer review of "PtrARF2.1 Is Involved in Regulation of Leaf Development and Lignin Biosynthesis in Poplar Trees"

_ijms, 2019, doi:10.3390/ijms20174141_

Round 1

Reviewer 1 Report

Review of IJMS-567760

Poplar PtrARF2.1 is involved in regulation of leaf development and lignin biosynthesis

Yongyao Fu, Papa Win, Huijuan Zhang, Chaofeng Li, Yun Shen, Fu He and Keming Luo

The authors used bioinformatics tools to identify six homologs of the Arabidopsis Auxin Response Factor 2 (ARF2) in Populus.trichocarpa. They then analyzed the expression of these 6 genes and found that PtrARF2.1 was predominantly expressed in leaf tissues and was significantly repressed by exogenous auxin treatment. PtrARF 2.2 and PtrARF 2.5 were also highly expressed in the studied tissues, whereas expression of the other three PtrARF2 family members was barely detected.

The authors chose to focus on PtrARF2.1. They constructed plants overexpressing this gene, and used RNAi technology to knock down its expression in other tissues. Plants overexpressing PtArf2.1 were slightly, but significantly, shorter than wild-type. In contrast, the plants in which PtrARF2.1 was inhibited by RNAi were significantly shorter than wild type. They also displayed altered leaf shape and lignin deposition. The authors used RNA seq to identify 2042 genes that were differentially regulated in PtrARF2.1 RNAi plants as compared to wild type. Of these, 74 encoded transcription factors and 24 were associated with lignin biosynthesis.

Overall, the gene expression and RNA-seq analyses provide clues as to the potential role of PtrARF2.1 in leaf development and indicate that it could be a useful target for plant breeding.

I would therefore like to know how stem and root development was affected in the RNAi and overexpression lines. Did the stems and roots also show the observed changes in lignin deposition? Altering lignin deposition in poplar stems could also be very useful, and should be easy to demonstrate.

I would also like to see the authors explain better why they chose to study PtrARF2.1. Since there are 39 ARF genes in Populus, why pick this one? They hint at it, but need to state it explicitly in their introduction, especially since PtArf2.2 shows high sequence similarity and has a very similar expression pattern to PtrARF2.1. I

Fig 3c shows that PtrARF2.1 transcripts were elevated in the OE lines. Does this result in elevated PtrARF2.1 protein levels? Why does both over- and under-expression of PtrARF2.1 stunt plant growth?

The authors state on lines 341-342 and again on line 401 that PtrARF2.1 may function as a negative regulator of auxin-responsive gene expression. It seems that the main support for this hypothesis comes from the fact that the GAL4:PtrARF2.1 fusion did not appear to activate transcription in yeast, and from the upregulation of many auxin-responsive genes in the RNAi lines. Since this would provide important insight into the way that PtrARF2.1 regulates leaf formation and lignin biosynthesis, this paper would be much stronger if they could actually demonstrate experimentally that PtrARF2.1 is a negative regulator of transcription.

The authors state on lines 91, 233 and 412 that PtrARF2.1 was predominantly expressed in poplar leaves. However, figure 2A shows that by far the highest levels of PtrARF2.1 transcripts were detected in roots, and the levels in leaves, stems and petioles were very similar. Please explain better or correct this statement.

The English is mostly understandable, but most sentences contain errors that require attention. I therefore recommend this paper be edited by a native English speaker.

Author Response

Reviewer 1

Review of IJMS-567760

Poplar PtrARF2.1 is involved in regulation of leaf development and lignin biosynthesis

Yongyao Fu, Papa Win, Huijuan Zhang, Chaofeng Li, Yun Shen, Fu He and Keming Luo

The authors used bioinformatics tools to identify six homologs of the Arabidopsis Auxin Response Factor 2 (ARF2) in Populus. trichocarpa. They then analyzed the expression of these 6 genes and found that PtrARF2.1 was predominantly expressed in leaf tissues and was significantly repressed by exogenous auxin treatment. PtrARF 2.2 and PtrARF 2.5 were also highly expressed in the studied tissues, whereas expression of the other three PtrARF2 family members was barely detected.

The authors chose to focus on PtrARF2.1. They constructed plants overexpressing this gene, and used RNAi technology to knock down its expression in other tissues. Plants overexpressing PtArf2.1 were slightly, but significantly, shorter than wild-type. In contrast, the plants in which PtrARF2.1 was inhibited by RNAi were significantly shorter than wild type. They also displayed altered leaf shape and lignin deposition. The authors used RNA seq to identify 2042 genes that were differentially regulated in PtrARF2.1 RNAi plants as compared to wild type. Of these, 74 encoded transcription factors and 24 were associated with lignin biosynthesis.

Overall, the gene expression and RNA-seq analyses provide clues as to the potential role ofPtrARF2.1 in leaf development and indicate that it could be a useful target for plant breeding.

I would therefore like to know how stem and root development was affected in the RNAi and overexpression lines. Did the stems and roots also show the observed changes in lignin deposition? Altering lignin deposition in poplar stems could also be very useful, and should be easy to demonstrate.

Response: Yes, we have checked the lignin deposition in the stems, and found an increased lignin deposition in the RNAi lines. Whereas we didn’t show these result here because we mainly focused on the effect of PtrARF2.1 on the leaf development.

I would also like to see the authors explain better why they chose to study PtrARF2.1. Since there are 39 ARF genes in Populus, why pick this one? They hint at it, but need to state it explicitly in their introduction, especially since PtArf2.2 shows high sequence similarity and has a very similar expression pattern to PtrARF2.1. 

Response: Firstly, it is previously reported that ARF2 is involved in the leaf development in model plants such as Arabidopsis and tomato. Six homologous genes of PtrARF2 were predicted in In poplar, showed 6 suggesting theirimportant roles. Secondly, as shown in Figure 2, both PtrARF2.1 and PtrARF2.2 were highly expressed in the leaves,but PtrARF2.2-RNAi lines showed relatively slight leaf changes in contrast to PtrARF2.1. Therefore we focused on the study of PtrARF2.1.

Fig 3c shows that PtrARF2.1 transcripts were elevated in the OE lines. Does this result in elevated PtrARF2.1 protein levels? Why does both over- and under-expression of PtrARF2.1 stunt plant growth?

Response: In fact, we only analyzed the transcript levels of PtrARF2.1 in transgenic lines. We speculated that PtrARF2.1 protein levels were also increased in the PtrARF2.1-OE lines. Over-expression of PtrARF2.1 in poplar didn’t show the reverse phenotypes as shown in PtrARF2.1-RNAi lines.  We think that ectopic expression of PtrARF2.1 in poplar caused stunt plant growth compared to wild-type plants.

The authors state on lines 341-342 and again on line 401 that PtrARF2.1 may function as a negative regulator of auxin-responsive gene expression. It seems that the main support for this hypothesis comes from the fact that the GAL4:PtrARF2.1 fusion did not appear to activate transcription in yeast, and from the upregulation of many auxin-responsive genes in the RNAi lines. Since this would provide important insight into the way that PtrARF2.1 regulates leaf formation and lignin biosynthesis, this paper would be much stronger if they could actually demonstrate experimentally that PtrARF2.1 is a negative regulator of transcription.

Response: ARFs with a glutamine-rich middle region (MR) usually act as activators of early auxin-responsive genes, while ARFs with proline- or/and serine-rich MRs or no specific amino acid-rich MRs as repressors. We analyzed the amino acids in the ARF domain and found that PtrARF2.1 had rich proline (P) and serine (S), and showed very highly homologous with the ARF domain of AtARF2, indicating that PtrARF2.1 might be a negative regulator of transcription.

The authors state on lines 91, 233 and 412 that PtrARF2.1 was predominantly expressed in poplar leaves. However, figure 2A shows that by far the highest levels of PtrARF2.1 transcripts were detected in roots, and the levels in leaves, stems and petioles were very similar. Please explain better or correct this statement.

Response: We have corrected the statement that PtrARF2.1 was predominantly expressed in poplar leaves in contrast to other PtrARF2 genes.

The English is mostly understandable, but most sentences contain errors that require attention. I therefore recommend this paper be edited by a native English speaker.

Response: Thanks for the suggestions of the reviewer. We have asked Dr. John Dixon (a native English speaker) from University of Georgia to improve our English.

Reviewer 2 Report

Fu and colleagues characterize one of the six ARF2 genes, ARF2.1, of Populus trichocarpa through the generation and study of gain and loss of function transgenic lines in the closely related species Populus tomentosa. The phenotypes of the lines make them conclude that PtrARF2.1 is involved in leaf development and lignin biosynthesis.

The research and its results are interesting and reinforce the role of ARF genes in important aspects of plant development, including woody species. However, the article need improvement before being considered to be publish.

General remarks:

I recommend authors deep review of the writing, and even the hire of professional translation service, since sometimes it is difficult to properly follow descriptions, procedures and explanations. In addition, there are quite typos all over the paper (line 32 “acids”, 55 “…D and Wagner D”, 66, “protin”, line 114 “confrimed”, line 130 “fragement”, line 317 “Tumefaciens”, line 226 “vitis” etc, etc) reflection of a not quite accurate review of the writing before submission.

Figures 1, 3, 4 and 7 should be bigger. These figures could be larger if they were widespread to fit the width of their corresponding figure legends.

When referred to Supplementary Data just indicate the table or figure with the S before the number (f.e: not "see Table S1 available as Supplementary Data", just "Table S1" is enough, for example in line 129 and many more).

Review by sections:

Abstract: Add some minimal information about the result of the yeast-one-hybrid experiment.

Introduction

Line 49. Add reference after CUC2.

Line 72 and 73. Revise font sizes.

Line 85 to 89. Revise punctuation and writing.

Line 94 PtrARF2a-RNAi, is the “a” a typo? Should be instead “2.1”?

Materials and methods

Line 152 2ΔΔCT, add reference of the method.

Line 173. Revise font sizes.

Results

3.1

Line 200. Change identified by confirmed.

Figure 1. It is very difficult, if not impossible, to read anything in figure 1A. Make bigger and use different colours to underline the different domains, for example, using the code of Figure S1.

3.2

There is a mistake when describing the organ expression of the different PtrARF2 genes: PtrARF2.5 in line 231 should be PtrARF2.6, and vice versa in line 232.

Figure 2B, could be moved to a new supplemental figure. I don´t consider necessary to perform a semiquantitative analysis when previously a quantitative analysis has been done.

Figure legend S1 is repeated in Figure legend S2, which in fact is missing. Please, change.

3.4

In my opinion, this is the core section of the article, but also the one that needs more clarification.

Since the RNAi lines are obtained in the P. tomentosa background, determination of the transcript levels of the different ARF2 genes, should be referred to the corresponding P. tomentosa paralog genes in all the section, isn´t it? For example: “expression analysis by qRT-PCR indicated that transcript level of PtrARF2.1 was specifically reduced in PtrARF2.1-RNAi and strikingly up-regulated in PtrARF2.1-OE lines, respectively (Fig. 3C).” When said that PtrARF2.1 is specifically reduced, it´s actually said that the expression of the paralog of PtrARF2.1 in P. tomentosa (that is PtoARF2.1) is reduced. If that is the case, name of the genes should change in this section and its corresponding supplemental material. Authors should consider this scenario, and explain and express it properly for the correct understanding of the readers.

It would be very interesting for the readers to see the leaf defects of the loss of function PtrARF2.1-RNAi line in figure 3D (a photograph at higher magnifications, or some additional ones, showing details would be fine).

Indicate in figure legend 3B how many plants were measured. In Figure 3C, the legend specifies "Quantitative analysis of RT-PCR of PtrARF2.1 transcripts in three independent lines of PtrARF2.1-RNAi", exactly as in the legend of Figure S4, where three bars are shown. However, I only see one bar in Figure 3C corresponding to an RNAi line. Please clarify. Additionally, there is no reference to the overexpression line in the legend of figure 3C. Please, include. Remove “*p-value<0.05” from the legend, since there is any bar with just one asterisk. Do the same in other figures of the paper when this happens.

In line 393 of the discussion sections, authors stand that “In previous reports, it was observed the ore14/arf2 Arabidopsis mutant exhibited enhanced sensitivity to auxin,…” Authors can check if the RNAi lines show also an enhanced sensitivity to auxin to reinforce that the phenotypes of the lines are related to an altered auxin signalling.

Finally, it would be interesting to discuss why the RNAi and overexpression lines show similar phenotype regarding size of plants (a height reduction). Moreover, please improve the wording of the sentence of line 286 "Whereas…", and also in the abstract line 18 (it´s not clear to me after reading this sentence if the overexpression lines are more similar to the wild type or to the RNAi lines).

3.5

Line 290. Change “Silence of” to “Silencing of”

Figure 4. Specify in line 308 if the wild-type in this case is Populus tomentosa. Mention in figure legend 4B the overexpression line as in figure 4A.

Section 3.6

Change the title of this section for something similar to “RNA-Seq transcriptome analysis of the PtrARF2.1 RNAi line”

When talking about “DEG” sounds much better “differentially expressed genes”, as authors do in line 162, than “different expression genes”, so please change all over the paper.

Line 321, what does the a of “PtrARF2a” means?

In Figure legend 5, the description of X and Y axe is flipped. Please, correct. Specify in line 328 if the wild-type in this case is Populus tomentosa.

Line 377, change “The dominant families” to “The families including more DEGs are….”, and remove “so on” at the end of the sequence since it´s to informal for a paper.

Line 338 WRKY instead of WAKY.

Figure 6. Change the legend of the Y axis to “Percentage of DEGs”.

Section 3.7

Please, try to rewrite the whole section, including the title. improve the title of this section.

Figure 7. It is not necessary to repeat the X labels in every plot of this figure, since all are referred to the same three lines (better add a little window with the three color code for these lines), and make the figure bigger.

Supplemental Material:

Figure S1. Add “respectively”, in the penultimate line after ARF domain.

Figure S2. Correct the legend.

Figure S4. Indicate what the asterisk means. In the ARF2.4 plot, isn´t statistically different the expression in ARF2.1-ARNi-1-2? Please check.

Supplementary Table S2. Specify in the legend what DW means, I mean “Dry Weight”.

Supplementary Table S5. Change “involving” to “involved” in the title.

Author Response

Reviewer 2

Comments and Suggestions for Authors

Fu and colleagues characterize one of the six ARF2 genes, ARF2.1, of Populus trichocarpa through the generation and study of gain and loss of function transgenic lines in the closely related species Populus tomentosa. The phenotypes of the lines make them conclude that PtrARF2.1 is involved in leaf development and lignin biosynthesis.

The research and its results are interesting and reinforce the role of ARF genes in important aspects of plant development, including woody species. However, the article need improvement before being considered to be publish.

General remarks:

I recommend authors deep review of the writing, and even the hire of professional translation service, since sometimes it is difficult to properly follow descriptions, procedures and explanations. In addition, there are quite typos all over the paper (line 32 “acids”, 55 “…D and Wagner D”, 66, “protin”, line 114 “confrimed”, line 130 “fragement”, line 317 “Tumefaciens”, line 226 “vitis” etc, etc) reflection of a not quite accurate review of the writing before submission.

Figures 1, 3, 4 and 7 should be bigger. These figures could be larger if they were widespread to fit the width of their corresponding figure legends.

When referred to Supplementary Data just indicate the table or figure with the S before the number (f.e: not "see Table S1 available as Supplementary Data", just "Table S1" is enough, for example in line 129 and many more).

Response: Thanks for the suggestions of the reviewer. We have changed all of these mistakes as suggested in General remarks.

Review by sections:

Abstract: Add some minimal information about the result of the yeast-one-hybrid experiment.

Response: Revised as suggested.

Introduction

Line 49. Add reference after CUC2.

Line 72 and 73. Revise font sizes.

Line 85 to 89. Revise punctuation and writing.

Line 94 PtrARF2a-RNAi, is the “a” a typo? Should be instead “2.1”?

 Response: We have revised as suggested.

Materials and methods

Line 152 2ΔΔCT, add reference of the method.

Line 173. Revise font sizes.

Response: We have revised as suggested by the reviewer.

Results

3.1

Line 200. Change identified by confirmed.

Figure 1. It is very difficult, if not impossible, to read anything in figure 1A. Make bigger and use different colours to underline the different domains, for example, using the code of Figure S1.

Response: According to the reviewer’s suggestion, the identical and similar amino acid residues are indicated by different colors, and the conserved B3 (DBD), ARF and Aux/IAA domains are underlined with red colours.

3.2

There is a mistake when describing the organ expression of the different PtrARF2 genes: PtrARF2.5 in line 231 should be PtrARF2.6, and vice versa in line 232.

Figure 2B, could be moved to a new supplemental figure. I don´t consider necessary to perform a semiquantitative analysis when previously a quantitative analysis has been done.

Figure legend S1 is repeated in Figure legend S2, which in fact is missing. Please, change.

Response:We have revised as suggested. In Figure 2B, we further detected transcript levels of PtrARF2 genes in poplar leaves by semiquantitative RT-PCR and found that transcript levels of only PtrARF2.1 and PtrARF2.2 were detected in the leaves. These results indicate that these two genes are involved in leaf development.

3.4

In my opinion, this is the core section of the article, but also the one that needs more clarification.

Since the RNAi lines are obtained in the P. tomentosa background, determination of the transcript levels of the different ARF2 genes, should be referred to the corresponding P. tomentosa paralog genes in all the section, isn´t it? For example: “expression analysis by qRT-PCR indicated that transcript level of PtrARF2.1 was specifically reduced in PtrARF2.1-RNAi and strikingly up-regulated in PtrARF2.1-OE lines, respectively (Fig. 3C).” When said that PtrARF2.1 is specifically reduced, it´s actually said that the expression of the paralog of PtrARF2.1 in P. tomentosa (that is PtoARF2.1) is reduced. If that is the case, name of the genes should change in this section and its corresponding supplemental material. Authors should consider this scenario, and explain and express it properly for the correct understanding of the readers.

Response: Yes, we agree to the reviewer’s opinions and have revised them as suggested.

It would be very interesting for the readers to see the leaf defects of the loss of function PtrARF2.1-RNAi line in figure 3D (a photograph at higher magnifications, or some additional ones, showing details would be fine).

Response: Please see Supplementary Figure S6.

Indicate in figure legend 3B how many plants were measured. In Figure 3C, the legend specifies "Quantitative analysis of RT-PCR of PtrARF2.1 transcripts in three independent lines of PtrARF2.1-RNAi", exactly as in the legend of Figure S4, where three bars are shown. However, I only see one bar in Figure 3C corresponding to an RNAi line. Please clarify Additionally, there is no reference to the overexpression line in the legend of figure 3C. Please, include. Remove “*p-value<0.05” from the legend, since there is any bar with just one asterisk. Do the same in other figures of the paper when this happens.

Response: In fact, we chose a representative line for displaying in Figure 3A and 3D. For quantitative analysis, three independent lines were used in these experiments. We have remove “*p-value<0.05”.

In line 393 of the discussion sections, authors stand that “In previous reports, it was observed the ore14/arf2 Arabidopsis mutant exhibited enhanced sensitivity to auxin,…” Authors can check if the RNAi lines show also an enhanced sensitivity to auxin to reinforce that the phenotypes of the lines are related to an altered auxin signalling.

Finally, it would be interesting to discuss why the RNAi and overexpression lines show similar phenotype regarding size of plants (a height reduction). Moreover, please improve the wording of the sentence of line 286 "Whereas…", and also in the abstract line 18 (it´s not clear to me after reading this sentence if the overexpression lines are more similar to the wild type or to the RNAi lines).

Response: We have revised these sentences as suggested. Over-expression of PtrARF2.1 in poplar didn’t show the reverse phenotypes as shown in PtrARF2.1-RNAi lines. We think that ectopic expression of PtrARF2.1 in poplar also caused stunt plant growth compared to wild-type plants.

3.5

Line 290. Change “Silence of” to “Silencing of”

Figure 4. Specify in line 308 if the wild-type in this case is Populus tomentosa. Mention in figure legend 4B the overexpression line as in figure 4A.

Response: We have revised it.

Section 3.6

Change the title of this section for something similar to “RNA-Seq transcriptome analysis of the PtrARF2.1 RNAi line”

When talking about “DEG” sounds much better “differentially expressed genes”, as authors do in line 162, than “different expression genes”, so please change all over the paper.

Line 321, what does the a of “PtrARF2a” means?

In Figure legend 5, the description of X and Y axe is flipped. Please, correct. Specify in line 328 if the wild-type in this case is Populus tomentosa.

Line 377, change “The dominant families” to “The families including more DEGs are….”, and remove “so on” at the end of the sequence since it´s to informal for a paper.

Line 338 WRKY instead of WAKY.

Figure 6. Change the legend of the Y axis to “Percentage of DEGs”.

Response: Thanks for the review’s suggestions and we have revised them.

Section 3.7

Please, try to rewrite the whole section, including the title. improve the title of this section.

Response: To make it more understanding, we have rewritten this section.

Figure 7. It is not necessary to repeat the X labels in every plot of this figure, since all are referred to the same three lines (better add a little window with the three color code for these lines), and make the figure bigger.

Response:Thanks for the suggestions of the reviewer. We have revised this figure.

Supplemental Material:

Figure S1. Add “respectively”, in the penultimate line after ARF domain.

Figure S2. Correct the legend.

Figure S4. Indicate what the asterisk means. In the ARF2.4 plot, isn´t statistically different the expression in ARF2.1-ARNi-1-2? Please check.

Supplementary Table S2. Specify in the legend what DW means, I mean “Dry Weight”.

Supplementary Table S5. Change “involving” to “involved” in the title.

Response: According to the comments of the reviewer, we have revised all of them.

Round 2

Reviewer 1 Report

Review of revised IJMS-567760

PtrARF2.1 is involved in regulation of leaf development and lignin biosynthesis in poplar

Yongyao Fu, Papa Win, Huijuan Zhang, Chaofeng Li, Yun Shen, Fu He and Keming Luo

The revision has addressed some of my concerns. For example, they now make a good case for focusing on PtrARF2.1. They also make a better case for why they think that it might be a transcriptional repressor, although the paper would be much stronger if they provided experimental evidence to support this statement.

However, some still need attention. For example, on line 232 they still state that that PtrARF2.1 was predominantly expressed in poplar leaves. However, figure 2A shows that by far the highest levels of PtrARF2.1 transcripts were detected in roots, and the levels in leaves, stems and petioles were very similar. Moreover, PtrARF2.2 shows a very similar pattern of expression. This needs to be corrected.

Similarly, they should at least briefly mention the effects on lignin deposition in the stems and roots, since this will also affect the dwarf phenotype.

I would also like them to address why overexpression of PtrARF2.1 also stunts growth in their discussion.

Finally, although the English is improved, many sentences still contain errors that require attention. For example, “figure” is misspelled as “figure” throughout the paper. I don’t have time to correct the entire paper, but just taking the introduction as an example:

Line 28 should be “…leaf development is a dynamic and complex process that responds to internal…”

Lines 31-32 should be “…Several phytohormones, including auxin (IAA), cytokinins (CK), gibberellins (GA) and jasmonic acid (JA), are involved in leaf initiation and development…”

Line 37: What does “these” refer to? I’m assuming auxin gradients, but please explain.

Line 43: “primordia” is plural, so “undergoes” should be “undergo”

Line 46 should be “…inhibit the growth of leaf lamina …”

Line 47: PAT should be defined the first time it is mentioned. Therefore, this line should be “…dictated by the polar auxin transport (PAT) system …”

Line 49 should be “…resulted in smooth leaves while …”

Lines 52-53 sound strange. Do they mean that mutation of these three importers results in serrated leaves? They have already described other ways to create serrated leaves, presumably without mutating these importers!

Lines 54-55 should be “…understanding auxin signal transduction (Ward and Estelle …”

Line 79 should be “…severe leaf curling…”

Line 85: delete “While”

Lines 91-92 should be “…slight changes in leaf phenotypes, transgenic plants carrying PtrARF2.1-RNAi displayed severe leaf phenotypes, such as irregular shapes and reduced size, while overexpression …”

Author Response

The revision has addressed some of my concerns. For example, they now make a good case for focusing on PtrARF2.1. They also make a better case for why they think that it might be a transcriptional repressor, although the paper would be much stronger if they provided experimental evidence to support this statement.

However, some still need attention. For example, on line 232 they still state that that PtrARF2.1 was predominantly expressed in poplar leaves. However, figure 2A shows that by far the highest levels of PtrARF2.1 transcripts were detected in roots, and the levels in leaves, stems and petioles were very similar. Moreover, PtrARF2.2 shows a very similar pattern of expression. This needs to be corrected.

Response: We have corrected the statement “PtrARF2.1 was predominantly expressed in poplar leaves” to “Expression profiles of PtrARF2 genes in different poplar tissues”.

Similarly, they should at least briefly mention the effects on lignin deposition in the stems and roots, since this will also affect the dwarf phenotype.

Response: we added the statement on line 322“Similar result was found in the stems of PtrARF2.1-RNAi lines (data not shown).”

I would also like them to address why overexpression of PtrARF2.1 also stunts growth in their discussion.

Response: We speculated that PtrARF2.1 might be involved in auxin homeostasis, resulting in repressing the development of transgenic plants.

Finally, although the English is improved, many sentences still contain errors that require attention. For example, “figure” is misspelled as “figure” throughout the paper. I don’t have time to correct the entire paper, but just taking the introduction as an example:

Line 28 should be “…leaf development is a dynamic and complex process that responds to internal…”

 Response: Revised as suggested.

Lines 31-32 should be “…Several phytohormones, including auxin (IAA), cytokinins (CK), gibberellins (GA) and jasmonic acid (JA), are involved in leaf initiation and development…”

 Response: Revised as suggested.

Line 37: What does “these” refer to? I’m assuming auxin gradients, but please explain.

 Response: Herein, “These” means “Leaf initiation”. The leaf initiation was generated by the affection of auxingradients.

Line 43: “primordia” is plural, so “undergoes” should be “undergo”

Response: Revised as suggested.

Line 46 should be “…inhibit the growth of leaf lamina …”

 Response: Revised as suggested.

Line 47: PAT should be defined the first time it is mentioned. Therefore, this line should be “…dictated by the polar auxin transport (PAT) system …”

 Response: Revised as suggested.

Line 49 should be “…resulted in smooth leaves while …”

 Response: Revised as suggested.

Lines 52-53 sound strange. Do they mean that mutation of these three importers results in serrated leaves? They have already described other ways to create serrated leaves, presumably without mutating these importers!

Response: No, they mean that mutation of these three importers results in the delay in leaf serration. We correct the statement “Mutation of three auxin importers (aux1/lax1/lax2) is required for delaying the serration growth”.

Lines 54-55 should be “…understanding auxin signal transduction (Ward and Estelle …”

 Response: Revised as suggested.

Line 79 should be “…severe leaf curling…”

Response: Revised as suggested.

Line 85: delete “While”

Response: Revised as suggested.

Lines 91-92 should be “…slight changes in leaf phenotypes, transgenic plants carrying PtrARF2.1-RNAi displayed severe leaf phenotypes, such as irregular shapes and reduced size, while overexpression …”

Response: Revised as suggested.